# Novel insights into RAGE signaling pathways during the progression of amyotrophic lateral sclerosis in RAGE-deficient SOD1 G93A mice

Natalia Nowicka[1]*, Kamila Zglejc-Waszak[1]*, Judyta Juranek[1], Agnieszka Korytko[1], Krzysztof Wąsowicz[2], Małgorzata Chmielewska-Krzesińska[2], Joanna Wojtkiewicz[1]

**1** Department of Human Physiology and Pathophysiology, Faculty of Medicine, Collegium Medium, University of Warmia and Mazury in Olsztyn, Olsztyn, Poland, **2** Department of Pathophysiology, Forensic Veterinary Medicine and Administration, Faculty of Veterinary Medicine, University of Warmia and Mazury in Olsztyn, Olsztyn, Poland

* kamila.zglejc@uwm.edu.pl (KZW); natalia.nowicka@uwm.edu.pl (NN)

**Data Availability Statement:** All relevant data are within the manuscript and its Supporting Information files as well as in the raw images.

## Abstract

Amyotrophic lateral sclerosis (ALS) is neurodegenerative disease characterized by a progressive loss of motor neurons resulting in paralysis and muscle atrophy. One of the most prospective hypothesis on the ALS pathogenesis suggests that excessive inflammation and advanced glycation end-products (AGEs) accumulation play a crucial role in the development of ALS in patients and SOD1 G93A mice. Hence, we may speculate that RAGE, receptor for advanced glycation end-products and its proinflammatory ligands such as: HMGB1, S100B and CML contribute to ALS pathogenesis. The aim of our studies was to decipher the role of RAGE as well as provide insight into RAGE signaling pathways during the progression of ALS in SOD1 G93A and RAGE-deficient SOD1 G93A mice. In our study, we observed alternations in molecular pattern of proinflammatory RAGE ligands during progression of disease in RAGE KO SOD1 G93A mice compared to SOD1 G93A mice. Moreover, we observed that the amount of beta actin (ACTB) as well as Glial fibrillary acidic protein (GFAP) was elevated in SOD1 G93A mice when compared to mice with deletion of RAGE. These data contributes to our understanding of implications of RAGE and its ligands in pathogenesis of ALS and highlight potential targeted therapeutic interventions at the early stage of this devastating disease. Moreover, inhibition of the molecular cross-talk between RAGE and its proinflammatory ligands may abolish neuroinflammation, gliosis and motor neuron damage in SOD1 G93A mice. Hence, we hypothesize that attenuated interaction of RAGE with its proinflammatory ligands may improve well-being and health status during ALS in SOD1 G93A mice. Therefore, we emphasize that the inhibition of RAGE signaling pathway may be a therapeutic target for neurodegenerative diseases.

## Introduction

The latest evidence on the role of the Receptor for Advanced Glycation End-products (RAGE) in the pathogenesis of neurodegenerative diseases revealed that RAGE plays a crucial role in

**Funding:** The work was financially supported by the National Science Centre (NCN), Poland (No. OPUS/2017/25/B/NZ4/00435). The publication fee was funded by the Minister of Science under "the Regional Initiative of Excellence Program".

**Competing interests:** Enter: The authors have declared that no competing interests exist.

amyotrophic lateral sclerosis (ALS) [1–6]. ALS is a motor neuron disorder characterized by progressive motor neuron atrophy as well as muscle deterioration, leading to premature death [1–6]. The crucial etiology of ALS still remains unknown; however, it has been determined that in many cases it is related to mutations in the gene encoding Cu/Zn superoxide dismutase (SOD1). Malfunctions in the SOD1 gene impairs cellular homeostasis of motor neurons, leading to the accumulation of Advanced Glycation End-products (AGE) in these cells [1–6]. We may speculate that during progression of ALS, AGEs trigger RAGE signaling pathway. Moreover, our earlier studies indicate that active RAGE-AGE pathway leads to neuroinflammation and thus neurodegeneration in ALS [7, 8]. However, the role of RAGE-AGE axis in progression of ALS is still not clear.

RAGE as an immunoglobulin receptor is a part of the innate immune system and may interact with proinflammatory ligands, such as: high mobility group box 1/amphoterin (HMGB1) and S calcium-binding protein B (S100B) and carboxymethyl-lysine-Advanced Glycation End-product (CML-AGE) [1–8]. Our previous studies revealed that RAGE and its proinflammatory ligands, such as: HMGB1, S100B as well as CML were elevated in the spinal cord of ALS mice and patients [3, 5, 6].

Furthermore, studies indicated that not only neuroinflammation proteins are observed during progression of ALS but also hyperactivation of glial cells were present in ALS patients and mice [1, 3, 8]. We observed elevated levels of astrocytes and microglia with a simultaneous disappearance of motor neurons in ALS spinal cord [1, 3, 8]. We explain that in ALS astrocytes may lose their physiological functions and release proinflammatory cytokines and chemokines and thus trigger motor neuron degeneration in the spinal cord [1, 3, 7, 9, 10]. Studies showed that hyperactive astrocytes were found in the ventral horn of ALS patients and mice [1, 9, 10]. Moreover, the results indicated that Glial Fibrillary Acidic Protein (GFAP) positive astrocytes revealed proinflammatory factors, such as: cytokines as well as RAGE proinflammatory ligands [1, 9, 10]. Elevated expression of proinflammatory proteins may trigger RAGE signaling pathway and thus motor neurons degeneration during progression of ALS [1–7].

Hence, the aim of the present study is to demonstrate the effect of RAGE signaling inhibition in SOD1 G93A transgenic mice. It is associated with decreased level of RAGE proinflammatory ligands as well as astrocyte markers in the lumbar spinal cord. We emphasize that RAGE plays a key role in ALS and may be a therapeutic target for neurodegenerative diseases.

## Materials and methods

### Animals

All procedures were performed according to the Local Ethical Committee of Experiments on Animals guidelines in Olsztyn, Poland (decision number 64/2018). Transgenic animals (B6. Cg-Tg(SOD1*G93A)1Gur/J) were purchased from The Jackson Laboratory (stock number 004435; Jackson Laboratories, Bar Harbor, ME, USA) and breed with RAGE knockout mice created by CRISPR-Cas9 method at the International Institute of Molecular and Cell Biology in Warsaw, Poland as described previously [6, 7]. The transgene copy number for SOD1 G93A and RAGE KO SOD1 G93A was examined as described (https://www.jax.org/strain/004435) [7]. To verify the knock-out of AGER (RAGE encoding gene), genotyping was performed using specific RAGE primers (Forward: TTGCTCTATGGGGTGAGACA, Reverse: GTAGACT CGGACTCGGTAG). Two RAGE products at 380 bp and 278 bp confirmed RAGE knock-out. Due to the estrogen protective effect, only male mice were used in the study. Male mice were housed in ventilated rooms under a 12:12 h light-dark period, lights on from 700 to 1900 h at 21˚C in with free access to food (Labofeed B standard) and water [6]. Experimental time points

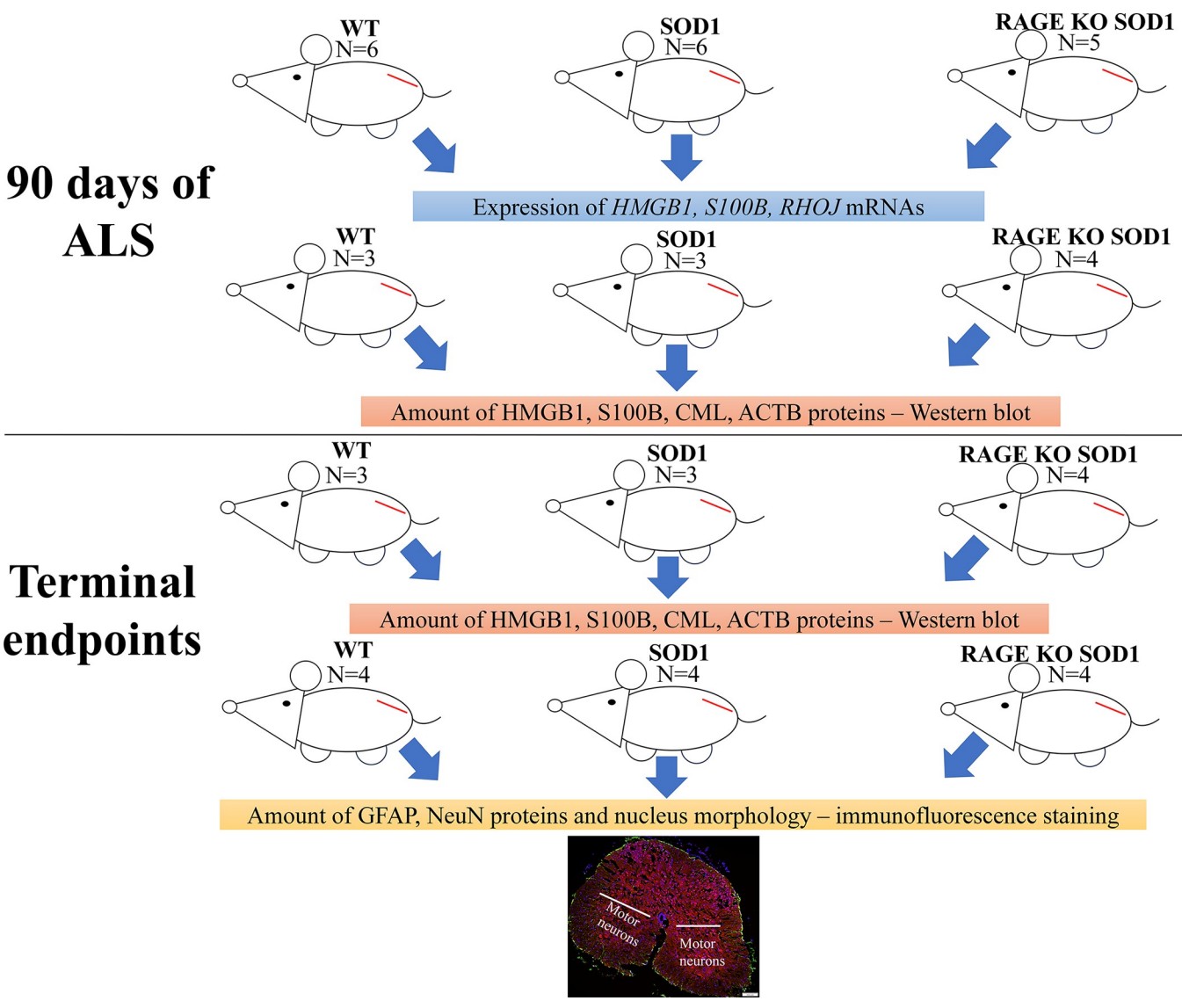

**Fig 1. Schematic diagram.** The diagram demonstrating subsequent stages of the experiment and showing the area of the spinal cord being examined; for the purpose of the study we examined lumbar motor spinal cord ventral horn lamina. The red line indicates the spinal cord. The figure is created by ourselves. Figure may be similar but not identical to the original image and is therefore for illustrative purposes only.

were defined at 90 and 150 days of age corresponding to the initial and final signs of motor deficits, respectively [6] (Fig 1). Mice (controls, SOD1 G93A, RAGE KO SOD1 G93A) from two time point groups, *i.e.* 90 and 150, were sacrificed independently at each of these time points regardless of their health status. As described in Nowicka et al. [6], the final time points were determined by 20% weight loss, or the mice's loss of self-righting ability (in mice SOD1 G93A). Mice were anesthetized with a mixture of ketamine (300 mg/kg) and xylazine (30 mg/kg) as described previously [3, 4, 6]. Next, mouse lumbar spinal cord tissue samples were frozen in liquid nitrogen and stored at −80°C for WB and RNA analysis as described previously in Nowicka et al. [6]. Samples for immunofluorescence staining were stored in appropriate buffers [3, 4].

**Table 1. Primers used for real-time PCR analysis.**

| Gene Symbol (Official) | Primer Sequences | Target Sequence Accession Number | Amplicon Length |
|---|---|---|---|
| *RHOJ* | F: 5′-TGTGTTTCTCATCTGCTTCTC -3′ | NM_023275.2 | 128 nt |
| | R:5′-GATCAATCTGGGTTCCGATG -3′ | | |
| *HMGB1* | F: 5′-GAGAAGGATATTGCTGCCTAC-3′ | U00431.1 | 160 nt |
| | R: 5′-CTTCATCTTCGTCTTCCTCTTC-3′ | | |
| *S100B* | F: 5′-GTCAGAACTGAAGGAGCTTATC-3′ | NM_009115.3 | 185 nt |
| | R: 5′-CATGTTCAAAGAACTCATGGC-3′ | | |
| *IPO8* | F: 5′-CTATGCTCTCGTTCAGTATGC-3′ | NM_001081113.1 | 173 nt |
| | R: 5′-GAGCCCACTTCTTACACTTC-3′ | | |
| *18S rRNA* | F: 5′-GGGAGCCTGAGAAACGGC -3′ | NR_003278.3 | 68 nt |
| | R: 5′-GGGTCGGGAGTGGGTAATTT -3′ | | |

### RNA and protein extraction

Total RNA (Wild type (WT, control) 90: n = 6; SOD1 90: n = 6; RAGE KO SOD1 90: n = 5) and protein (WT 90 and 150: n = 3; SOD1 90 and 150: n = 3; RAGE KO SOD1 90 and 150: n = 4) from the same lumbar spinal cord neuromere in each animal studied was isolated using the AllPrep DNA/RNA/Protein Mini Kit (Qiagen, Hilden, Germany) according to the manufacturer's protocol. The quantity and quality of RNA was verified spectrophotometrically using Thermo Scientific™ NanoDrop ™ 2000/2000c Spectrophotometer (Thermo Fisher Scientific, Waltham, Massachusetts, USA). The protein concentration was determined using Direct Detect® Infrared Spectrometer for Total Protein Quantitation (Merck Millipore, Darmstadt, Germany).

### Reverse transcription and quantitative PCR

One microgram of RNA was transcribed into cDNA by The QuantiNova Reverse Transcription Kit (Qiagen, Hilden, Germany) according to the manufacturer's instructions. The obtained cDNA was used for quantitative PCR analysis (Table 1). The expressions of mRNA transcripts of genes encoding *HMGB1*, *S100*B [6], ras homolog family member J (*RHOJ)* [10] were investigated in duplicate by SYBR® Green PCR Master Mix (Qiagen, Hilden, Germany) and a LightCycler® 480 Instrument II (Roche Molecular Systems, Inc., Switzerland). The initial denaturation at 95°C for 10 min was followed by 40 cycles of denaturation at 95°C for 5 s, primer annealing at 62°C for 10 s, and elongation at 72°C for 10 second. All amplifications were followed by dissociation curve analysis of the amplified products. The relative expression of mRNAs was calculated by $\Delta\Delta$Ct method and normalized using the geometric mean of expression levels of housekeeping genes, *i.e.* importin 8 (*IPO8*) [6] and 18S ribosomal RNA (*18S rRNA)* [10, 11].

### Western blot analysis

Proteins (40μg, WT 90 and 150: n = 3; SOD1 90 and 150: n = 3; RAGE KO SOD1 90 and 150: n = 4) were separated on 15-well 4–15% Mini-PROTEAN® TGX™ Precast Protein Gels (Bio-Rad, Hercules, CA, USA) and transferred onto nitrocellulose membrane using semi-dry system (Trans-Blot Turbo Transfer System, Bio-Rad). Prior to antibody incubation the blotting membrane was blocked in EveryBlot Blocking Buffer (Bio-Rad) for 5 min at room temperature (RT). Primary antibody solutions were diluted in SignalBoost™ Immunoreaction Enhancer solution for primary antibodies (Merck Millipore, Darmstadt, Germany) and left for overnight incubation at 4°C. The list of all primary antibodies is presented in Table 2. After four washes

Table 2. Antibodies used for Western Blot analysis and immunofluoresence staining.

| Primary Antibodies | | | | |
| --- | --- | --- | --- | --- |
| Antigen | Code | Species | Working Dilution | Supplier |
| NeuN | ab177487 | Rabbit | 1:100 | Abcam, Cambridge, UK |
| HMGB1 | ab18256 | | 1:1 000 | |
| S100b | ab52642 | | 1:1 000 | |
| CML | ab27684 | | 1:5 000 | |
| ACTB | ab6276 | | 1: 1 000 | |
| GFAP | 173006 | Chicken | 1:500 | Synaptic System |
| Secondary Antibody | | | | |
| Reagents | Code | | Working Dilution | Supplier |
| StarBright Blue 700 Goat Anti-Rabbit IgG | 12004158 | | 1:10 000 | BioRad Hercules, CA, USA |
| StarBright Blue 700 Goat Anti-Rabbit IgG | 12004162 | | 1:10 000 | |
| IgG (H + L), Alexa Fluor Plus 594 | A11039 | | 1: 2 000 | ThermoFisher |
| IgY (H + L), Alexa Fluor 488 | A-32740 | | | |
| Fluoroshield™ with DAPI | F6057-20ML | | 3–4 drops of mounting medium directly on top of the specimen | Sigma-Aldrich, St. Louis, MO, USA |

in PBS with 0.1% Tween-20, the membrane was incubated for one hour at RT with the corresponding fluorescence-labeled secondary antibody diluted in SignalBoost™ Immunoreaction Enhancer solution for secondary antibodies (Merck Millipore, Darmstadt, Germany). The bands were visualized with ChemiDoc Imaging Systems (Bio-Rad Hercules, CA, USA). Images were quantified densitometrically with ImageJ Software 1.50i (Wayne Rasband, National Institutes of Health, Bethesda, MD, USA) and compared between and within each group—control versus experimental, after normalization to the total amount of protein loaded in the gel. Detailed description in Nowicka et al. [6].

## Immunofluorescence

The nine-micron sections of lumbar spinal cord tissues (n = 4 per groups) were subjected into routine immunofluorescence technique according to the method described previously by [12]. Sections were dried at room temperature (RT) for 45 min and blocked with a solution containing goat serum for 1 hour at room temperature. Afterwards, the sections were rinsed three times by PBS and incubated overnight at RT in humid chamber with combinations of appropriate antibodies against GFAP (glial fibrillary acidic protein) and NeuN (neuronal nuclei). The next day, the tissues were rinsed by PBS and incubated with species-specific secondary antibodies detailed in Table 2. Negative control confirmed antibody specificity (S1 Fig).

DAPI is a dye that can be used as a tool to visualize nuclear morphological changes. Slides with tissues were mounted with DAPI (4′,6-diamidino-2-phenylindole, Sigma-Aldrich, USA) to visualize nuclei of stained cells and subsequently, examined under fluorescent microscope. Cells with damaged nuclei stained with DAPI may show noticeable nuclear blebbing (fragmented nuclei) which may help differentiate morphological changes compared to cells without nuclear changes. Morphological changes were defined as the proportion of noticeable nuclear blebbing to normal nuclei in each group.

Immunostained tissues were photographed using a fluorescence microscope (Olympus IX83) and analyzed with NIH open source Image J software. Subsequently, the areas of immunofluorescent staining were measured in four similar regions (ROI) of interest for each tissue slice (ventral horn). Four technical replicates were used per each of four biological replicates within one studied group. From these values, the mean area of fluorescent signal, representing

the amount of GFAP and NeuN proteins and thus positive cells in each group, was calculated as described in Zglejc-Waszak et al. [10].

## Statistical analysis

All analyses were performed using GraphPad Prism (San Diego, CA, USA) as described previously [6, 10, 13]. Data are presented as means ± S.E.M. The normality and lognormality test (Shapiro-Wilk test) was performed before the statistical analysis. The mRNA expression of *HMGB1*, *S100B* and *RHOJ* was compared using Kruskal–Wallis one-way analysis of variance. The amount of proteins in WB as well as immunofluorescence were compared by one-way ANOVA with Tuckey's post-hoc test or non-parametric equivalent, *i.e.* Kruskal–Wallis test. $P \leq 0.05$ value was considered as statistical significance.

## Results

Three groups of mice (wild type, SOD1 and RAGE KO SOD1, Fig 1) were compared in terms of coping with the disease. Mice from all groups were constantly monitored as described in Nowicka et al. [6].

### Gene expression levels of *HMGB1*, *S100B* as well as *RHOJ*

At 90 day time point no significant changes were observed for *HMGB1*, *S100B* and *RHOJ* mRNAs (P > 0.05, Fig 2A–2C). However, the highest expression of *HMGB1* as well as *S100B* mRNAs was observed in spinal cord harvested from SOD1 mice (Fig 2A) and the lowest expression of mentioned mRNAs was observed in RAGE KO mice (Fig 2A and 2B), indicating that the deletion of RAGE may reduce, without statistical significance (P > 0.05), the expression of major proinflammatory RAGE ligands in the spinal cord of SOD1 G93A mice. GTPAses may play a central role in motor neuron damage during ALS. The activation of

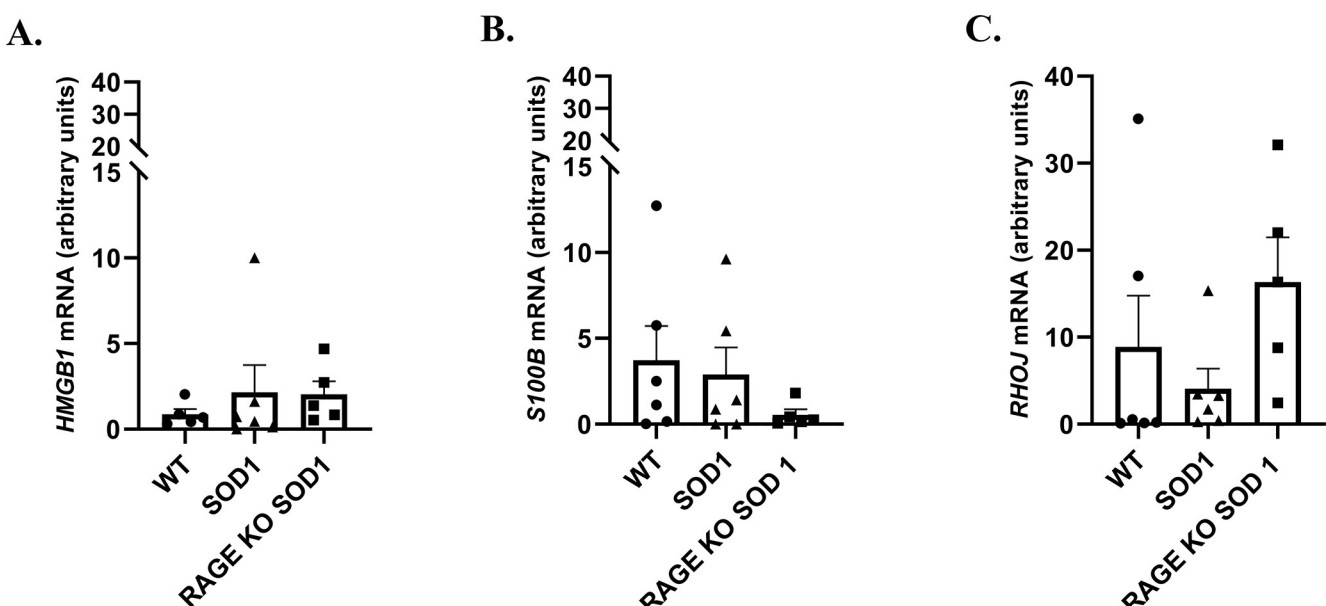

**Fig 2.** The relative expression of *HMGB1* (A), *S100B* (B) and *RHOJ* (C) mRNAs in the lumbar spinal cord harvested from SOD1 transgenic mice with congenital, ALS RAGE knockout SOD1 transgenic mice with congenital ALS, and wild type mice at 90 days' time points. Data are presented as means ± S.E.M. relative to the geometric mean of the expression level of *IPO8* and *18SrRNA*. Abbreviation: WT–wild type group, n = 6; SOD1 group, n = 6; RAGE KO SOD1, n = 5.

GTPases is altered in the G93A mutant hSOD1 mouse model of ALS [14, 15]. Moreover, RHOJ as another GTPase may activate RAGE signalling pathway in neurodegenerative diseases. Nevertheless, GTPAses may interact also with cytoskeleton proteins and thus play role in polymerization of actin cytoskeleton in cells. RHOJ, as a member of Rho proteins, regulates the dynamic assembly of cytoskeletal components and thus may influence synaptic degeneration in ALS. However, the role of GTPAses as well as RHOJ in the regulation of motor neuron survival is unclear [14]. Hence, we investigated whether genetic deletion of RAGE in SOD1 G93A mice altered the expression of *RHOJ* mRNA in the lumbar spinal cord. We identified a reduction, without statistical significance (P > 0.05), in mRNA expression of *RHOJ* in spinal cord of SOD1 G93A mice when compared with mice RAGE KO SOD1 (Fig 2C).

## Immunoblot levels of HMGB1, S100B, CML as well as ACTB proteins in mouse lumbar spinal cord

We did not observe significant changes in expression of inflammatory proteins as well as in ACTB in our groups of animals at the terminal time point (P ≥ 0.05; Fig 3A–3D). Nevertheless, the highest amount of HMGB1 as well as ACTB was observed in spinal cord harvested from SOD1 G93A mice (Fig 3A and 3D).

Next, we investigated whether time point of experiment and RAGE deletion in SOD1 G93A mice altered the amount of HMGB1, S100B, CML as well as ACTB proteins in lumbar spinal cord (S1 Raw images). Our results showed alternations in proteins abundant in RAGE knockout SOD1 G93A mice over the duration of the disease/experiment (P ≤ 0.05; Fig 4A–4C). Proinflammatory ligands of RAGE as HMGB1 and S100B were significantly decreased in terminal group compared to 90 day time points (Fig 4A and 4B; P ≤ 0.001, P ≤ 0.05; respectively). However, we observed increasing trend of CML and decreasing trend of ACTB at the terminal point in RAGE knockout SOD1 G93A mice (Figs 3C and 4C). This suggests that while RAGE deficiency and duration of illness did not impact CML and ACTB levels, the amount of proinflammatory ligands, HMGB1 and S100B was reduced in affected SOD1 G93A spinal cord (Fig 4A–4C).

## Glia and neuronal immunostaining in mouse lumbar spinal cord

We examined the percentage of damaged nuclei, microglia and astrocyte accumulation in ventral horn of murine spinal cord. Numerous studies demonstrated motor neuron damage as well as elevated level of microglia and astrocytes in ALS lumbar spinal cord. We first studied whether the deletion of RAGE in SOD1 G93A mice affect the structure of neuronal nuclei in murine lumbar spinal cord. Results showed elevated level of nuclear pathological changes in SOD1 mice compare to wild type group (Fig 5A and 5B; P ≤ 0.05) however we did not see any changes in RAGE KO SOD1 group. Moreover, we did not find any differences in nuclear morphology between SOD1 G93A mice with genetic deletion of RAGE and control group (Fig 5A and 5B, P ≥ 0.05). Next, we investigated the level of GFAP (marker of astrocytes) in ventral horn of lumbar spinal cord at the terminal stage of the disease in all three groups of animals. Similar to nuclear damage, GFAP protein was also increased in SOD1 G93A mice when compared with RAGE KO SOD1 G93A and control mice (Fig 5A and 5C; P ≤ 0.0001; P ≤ 0.001). Surprisingly, the level of GFAP was decreased in RAGE KO SOD1 G93A compared to control mice (Fig 5A and 5C, P ≤ 0.001). Thus, the lowest level of astrocyte was observed in ventral horn of lumbar spinal cord harvested from RAGE KO SOD1 G93A mice. Contrary to the observed alternations in glial marker expression, the immunoreactive area of NeuN as a Neuronal Nuclear Antigen and Neuron Differentiation Marker, did not change between RAGE KO SOD1 G93A and SOD1 G93A mice (Fig 5A–5C, P ≥ 0.05). Moreover, we observed that

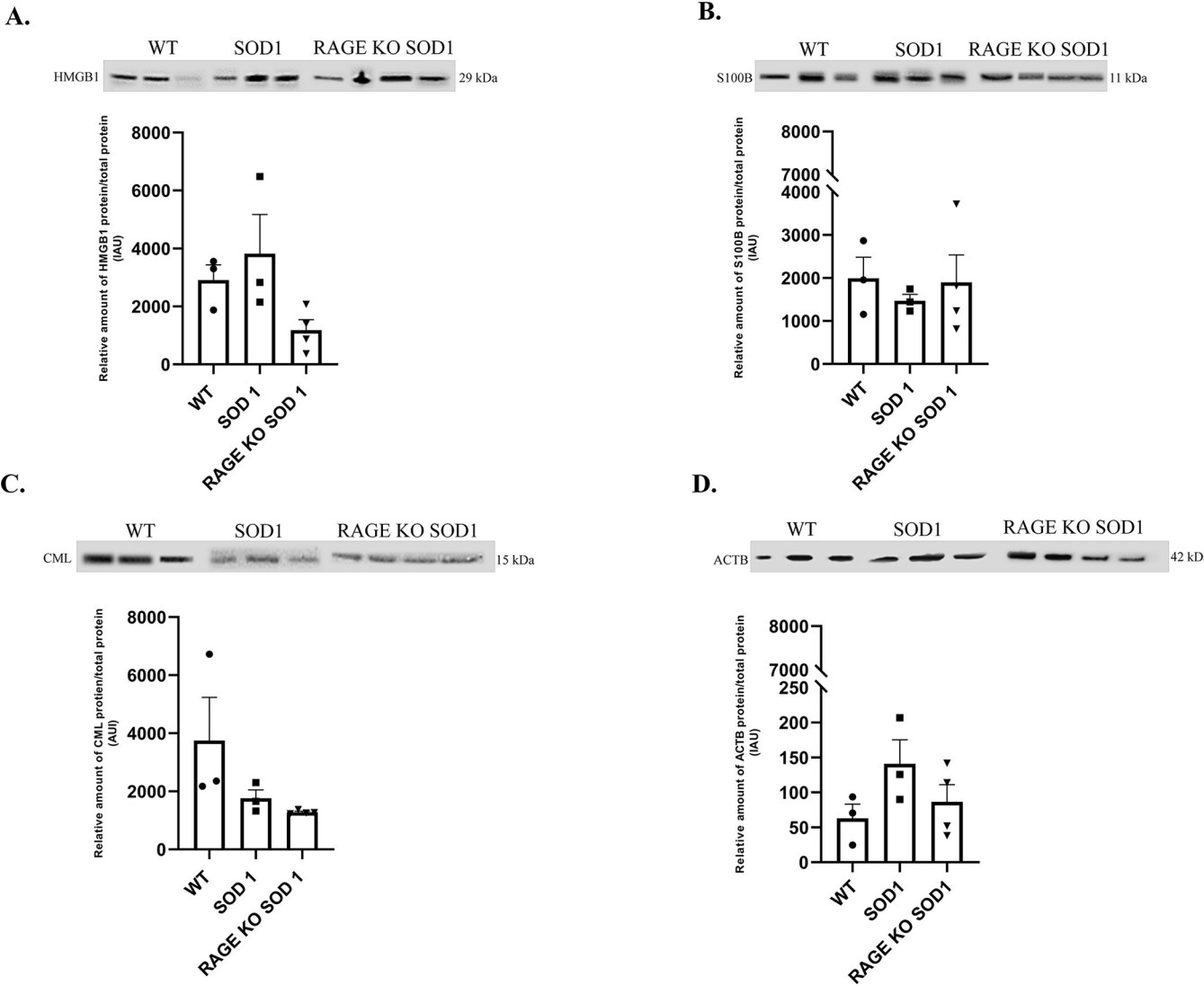

**Fig 3.** The relative amount of proteins: HMGB1 (A), S100B (B), CML (C), ACTB (D) in the lumbar spinal cord harvested from SOD1 transgenic mice with congenital ALS, RAGE knockout SOD1 transgenic mice with congenital ALS, and wild type mice at terminal time point quantified by western blot. Figure panels are indicated above the corresponding western blot set. Data are presented as means ± S.E.M. There are no statistical differences in relative amount of proteins. Abbreviation: WT–wild type group, n = 3; SOD1 transgenic mice with congenital ALS, n = 3; RAGE KO SOD1 –RAGE knockout SOD1 transgenic mice with congenital ALS, n = 4, at terminal endpoints.

NeuN is expressed in GFAP-positive cells of lumbar spinal cord (Fig 5A). Darlington et al. [15] compared widespread immunoreactivity for NeuN that would correspond to our observation. Our results showed that genetic deletion of RAGE reduces astrocyte accumulation in the spinal cord of SOD1 G93A mice (Fig 5A–5D).

## Discussion

Our studies indicated that inhibition/lack of RAGE signaling pathway in spinal cord of ALS mice exerts an anti-inflammatory effect. Moreover, genetic deletion of RAGE reduces astrocyte accumulation in the spinal cord of SOD1 G93A mice. This effect was associated with a reduction expression of proinflammatory RAGE ligands and less pronounced damage of nuclei in lumbar spinal cord of ALS mice.

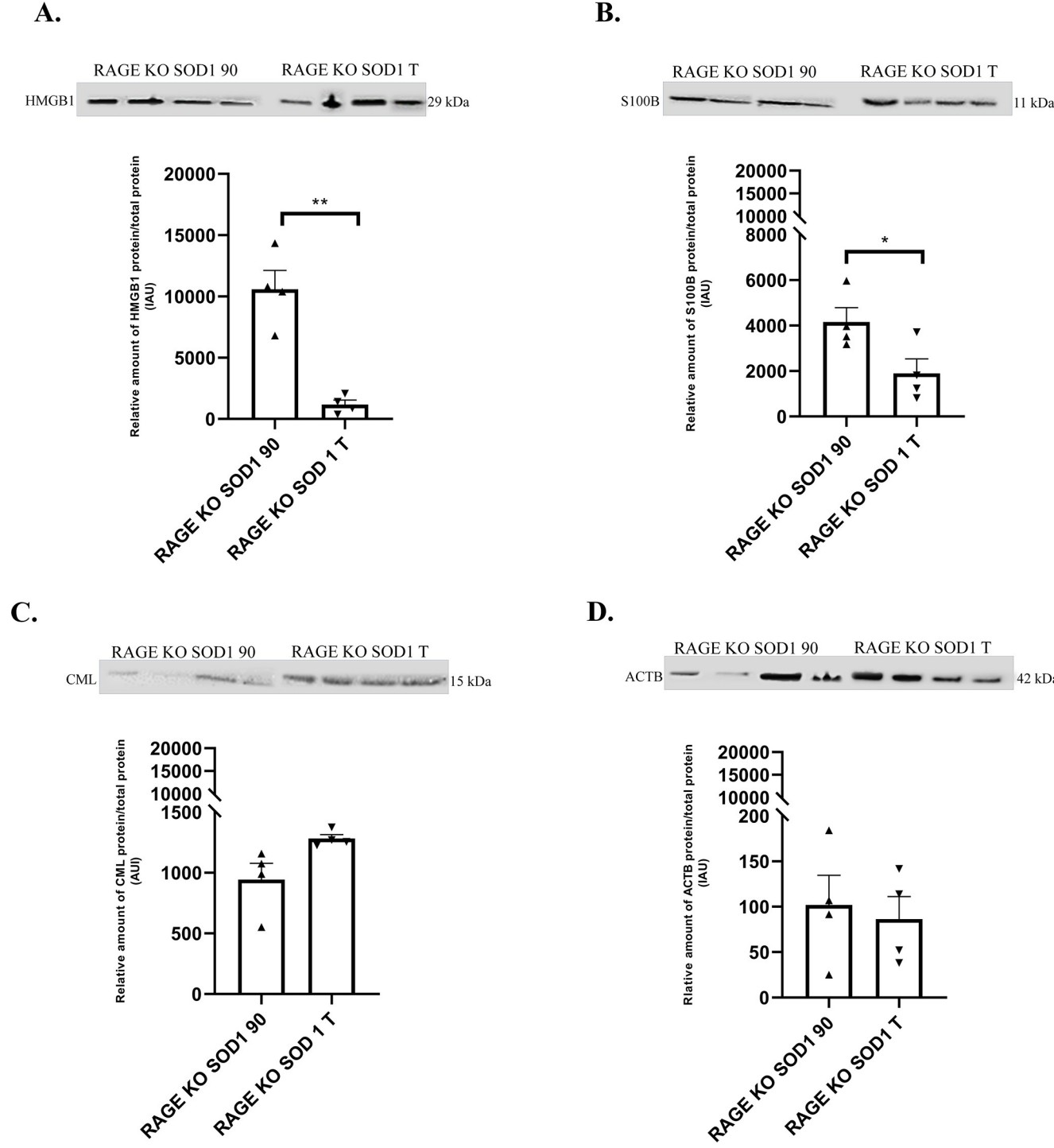

**Fig 4.** The relative amount of proteins: HMGB1 (A), S100B(B), CML(C), ACTB (D) in the lumbar spinal cord harvested from SOD1 transgenic mice with congenital ALS, RAGE knockout SOD1 transgenic mice with congenital ALS, and wild type mice at 90 and terminal time points quantified by western blot. Figure panels are indicated above the corresponding western blot set. Data are presented as means ± S.E.M. Statistical differences in relative amount of protein respectively * 0.01 ≤ P ≤ 0.05; ** 0.001 ≤ P≤ 0.01. Abbreviation: RAGE KO SOD1 90 –RAGE knockout SOD1 transgenic mice with congenital ALS n = 4 at 90 time point, RAGE KO SOD1 T–RAGE knock out SOD1 transgenic mice with congenital ALS n = 4, at terminal endpoints.

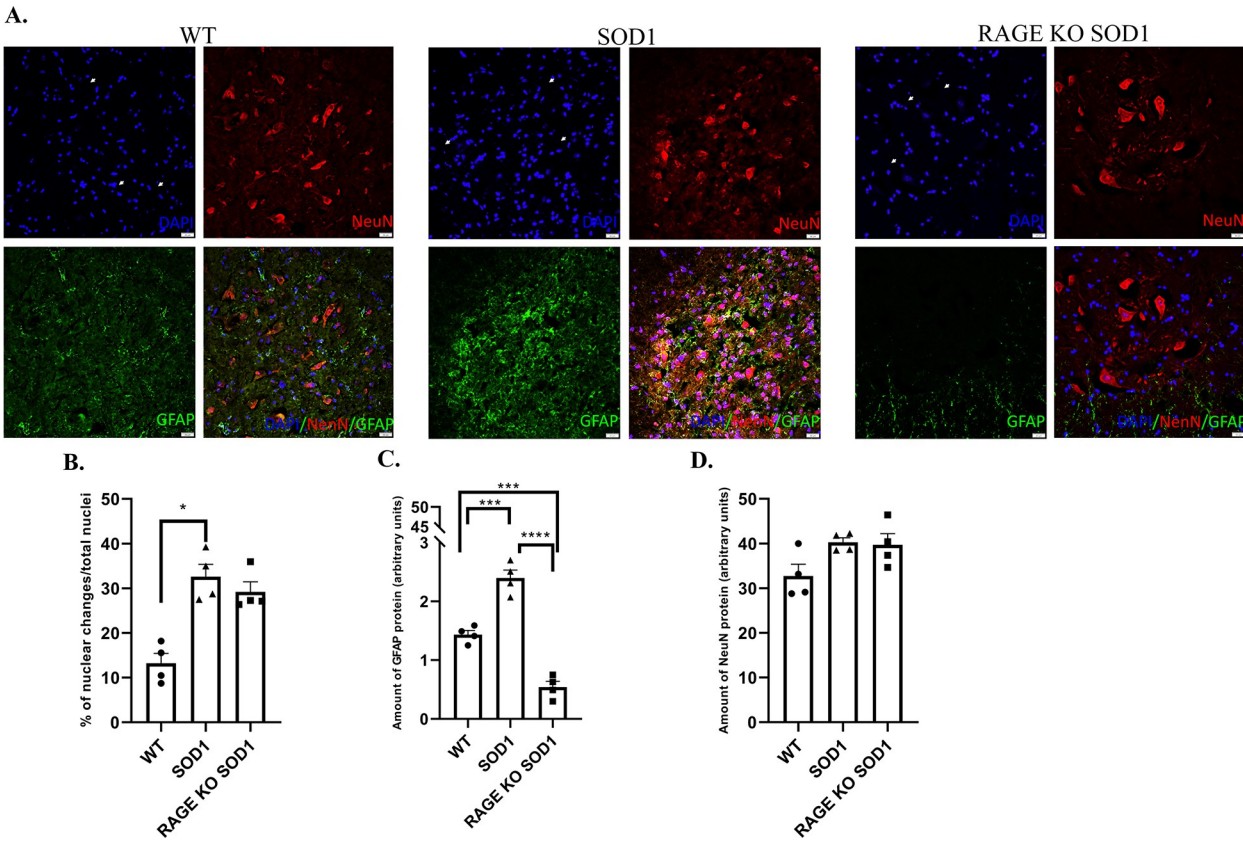

**Fig 5. Neuronal count and GFAP as well as NeuN expression in terminal-stage SOD1 transgenic mouse ventral horn of spinal cord.** A. DAPI positive nuclei (blue), GFAP protein showed (green), NeuN neurons showed (red). Co-localization (yellow staining) of GFAP-NeuN in mice spinal cord. Images were taken under 20× objective with 0.7 numerical aperture (20× /0.7). Scale bar = 20 μm. For the purpose of the study we examined lumbar motor spinal cord ventral horn lamina. B. Effect of RAGE knockout on nuclear changes and assess apoptosis in ALS mice. Arrows indicated alternations in nuclei. C. Genetic deletion of RAGE effects on glial marker (GFAP protein, green colour) in the spinal cord of SOD1 mice. D. SOD1 mice lacking RAGE have the same level of NeuN in spinal cord (motor neurons–red). Statistical differences in relative amount of protein respectively * $0.01 \leq P \leq 0.05$; ** $0.001 \leq P \leq 0.01$.

It is well documented that RAGE signaling pathway is involved in ALS progression [1, 3, 5, 6]. Our previous studies revealed elevated level of RAGE in the lumbar spinal cord of SOD1 G93A mice [1, 3–6]. Moreover, we observed altered expression of HMGB1, S100B as well as CML in mice with ALS [1, 3–6]. Our current results support the prior hypothesis that inhibition/lack of RAGE signaling pathway in ALS mice exerts an anti-inflammatory effect (Fig 6).

We revealed that genetic deletion of RAGE may reduce expression of *HMGB1* and *S100B* mRNAs in the lumbar spinal cord of SOD1 G93A mice at 90 days of disease. However, our previous studies indicated that the highest expression of *HMGB1* mRNA in spinal cord harvested from mice at 60 days of disease [6]. Lee et al. [15] indicated that SOD1 G93A mice lacking RAGE have decreased spinal cord levels of HMGB1 and other key proinflammatory markers. Nevertheless, we observed the elevated level of *S100B* mRNA in spinal cord harvested from mice at 90 days of disease [6]. These data revealed that RAGE signaling pathway plays an inflammatory role in the SOD1 G93A mice. Proinflammatory RAGE ligands, such as HMGB1, S100B as well as CML, bind to RAGE and trigger production of inflammatory cytokines, leading to motor neuronal loss [1–6, 16]. However, further studies are necessary to explain the phenomenon that SOD1 G93A mice lacking RAGE have decreased spinal cord level of key inflammatory markers (Fig 6).Our studies revealed that the highest amount of HMGB1

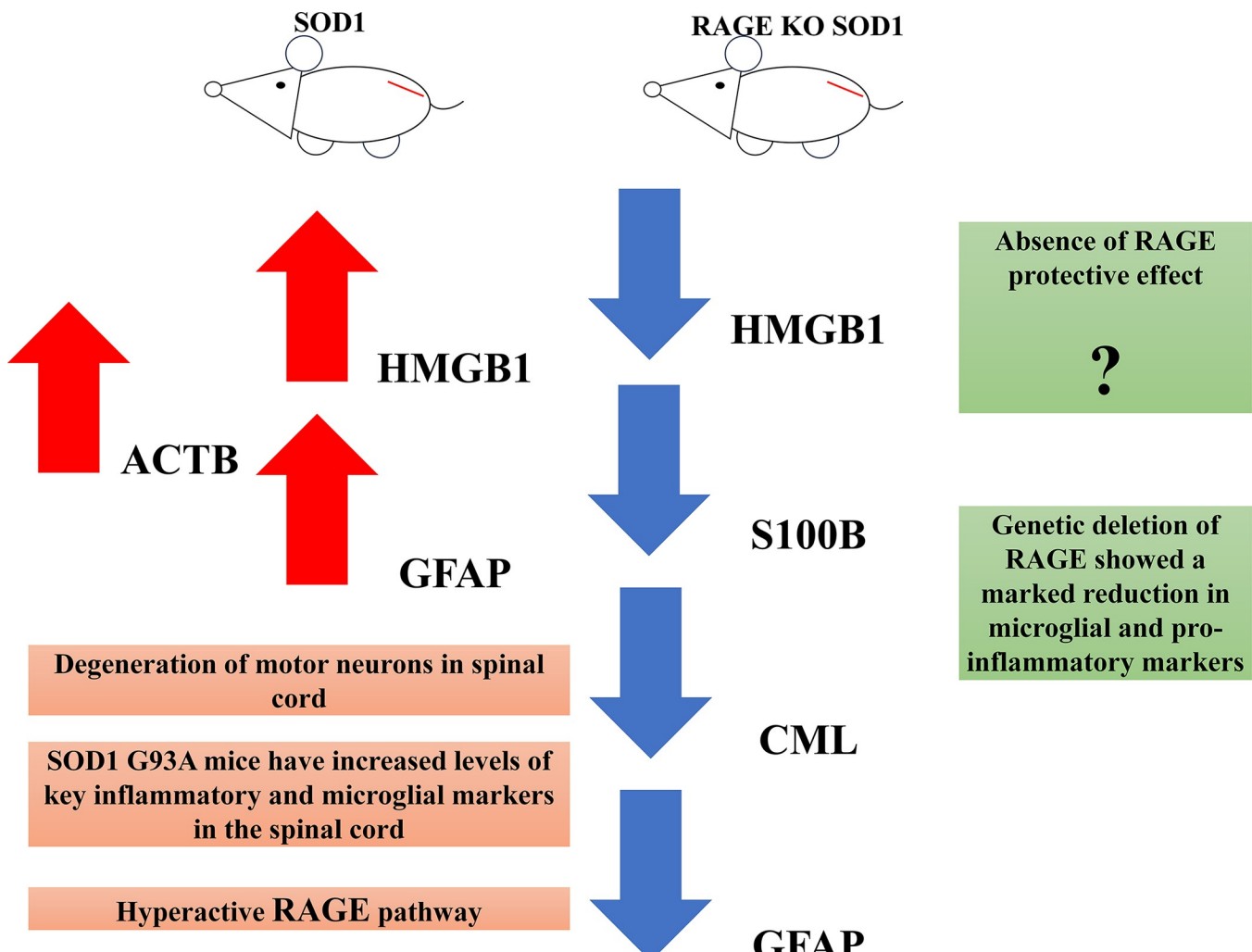

**Fig 6. Summary of the study.** Genetic deletion of RAGE reduces glial marker and major inflammatory proteins in the spinal cord of SOD1 G93A mice. The figure is created by ourselves. Figure may be similar but not identical to the original image and is therefore for illustrative purposes only.

protein was in SOD1 mice. Our data corroborate the premise that enhanced RAGE signaling plays a key role in neuronal inflamed and thus ALS progression [6]. Previous studies revealed the highest activity of RAGE signaling pathway at the terminal stage of ALS compared to control [6]. However, our previous studies revealed the highest amount of HMGB1 as well as S100B at the beginning of the study [6]. Therefore, we hypothesize that enhanced HMGB1 activity in SOD1 G93A mice and thus RAGE signaling pathway could drive neuroinflammatory responses that trigger motor neuron loss in spinal cord [17–20].

Schmidt research team [21, 22] found ten small molecules that inhibit ligand-stimulated RAGE signal transduction. Their studies revealed that interactions these small molecules with RAGE may decrease inflammation score during cardiac ischemia-reperfusion injury [21, 22]. Hence, we may hypothesize that genetic deletion of RAGE may attenuate inflammation in spinal cord harvested from SOD1 G93A mice and thus motor neuron destroyed in ALS SOD1 G93A mice. Lee et al. [18] revealed that therapeutic blockade of HMGB1 attenuates interaction with RAGE reduces early motor deficits, but not survival in SOD1 G93A mouse model of ALS. Thus, we may speculate that the attenuated activity of HMGB1 may be beneficial during the

progression of ALS. We observed decreased amount HMGB1, S100B as well as CML (proinflammatory RAGE ligands) in spinal cord of RAGE KO SOD1 G93A mice [1, 3, 5–7, 15, 16, 23]. Moreover, the studies indicated that RAGE KO SOD1 G93A mice are characterized by extended survival time, improved motor performance and slowed disease progression [18]. Absence of RAGE reduces inflammation, improves the quality of life and extends survival in the SOD1 G93A mouse. Interestingly, the genetic deletion of RAGE during the duration of the disease resulted in a deepening of attenuated downregulation of RAGE signaling pathway components [15, 23]. Hence, it is possible that the deletion of RAGE could reduce inflammation and motor neuron damage. Alterations in the pattern of protein amount of proinflammatory RAGE ligands during ALS in RAGE KO SOD1 G93A mice may trigger regeneration of motor neurons in lumbar spinal cord [20]. However, further studies are needed to explain this phenomenon (Fig 6).

New studies have revealed that anti-ACTB antibody may be a new biomarker for progression of ALS and may mark the disease severity [24]. ACTB is the most important key component of the cytoskeleton of cells, including neuronal cells. Beta actin plays a crucial role in controls cell growth, movement as well as migration. Moreover, it controls formation of new synaptic connections as well as a new branch formation of neurons [25, 26]. Our data indicated that increased amount of ACTB was present in SOD1 G93A mice. Therefore, we may speculate that the amount of protein correlated with the progression of the disease, indicating that the formation of new synaptic connections as well as new branch formation of neurons were malfunctioning during the progression of the neurodegenerative disease [16, 25, 26]. Our data confirm that ACTB may be a new biomarker of ALS in patients and also in SOD1 G93A mice. However, mechanism of the elevated amount of ACTB in SOD1 G93A mice is unknow. We can speculate that during the progression of ALS when motor neurons are damaged, ACTB may be excessively released and also reconstructed resulting in the increased amount of this protein in spinal cord harvested from SOD1 G93A mice as well as in serum of patients with ALS [24–26]. Nevertheless, the question of the proportionality of ACTB levels in serum and spinal cord harvested from ALS mice in not straightforward. The serum/blood analysis may reflect certain alterations in gene expression and thus protein amount within the spinal cord [24–26]. However, further studies are necessary to clarify this phenomenon.

Further, our results showed that in ALS mice the percentage of damaged nuclei is higher than in control group. However, we did not observe alternations in nuclear number between RAGE KO SOD1 G93A mice and control animals. DNA damage is a mechanism of neurodegeneration in ALS and contributes to astrocyte toxicity [9, 27]. Although deletion of RAGE did not affect neuronal nuclear morphology in lumbar spinal cord, surprisingly, GFAP protein expression was altered. We observed decreased level of GFAP, astrocyte marker, in RAGE KO SOD1 G93A mice compared to SOD1 G93A and control mice. It is well documented that the increased level of GFAP accelerate astroglial activation and gliosis during neurodegeneration [9, 17, 28–34]. Lee and co-workers [7] indicated that astorcyte marker within lumbar spinal cord was not altered. Thus, we may speculate that during the progression of ALS different pathological mechanisms are active [20, 21, 27–29]. Our data showed the same level of NeuN in RAGE KO SOD1 G93A mice and control animals. Moreover, we did not find differences in the amount of neuronal marker between RAGE KO SOD1 G93A and SOD1 G93A mice. It is therefore likely that amelioration of spinal cord gliosis following the deletion of RAGE in ALS mice led to decreased motor neuron loss, subsequently reducing neurodegeneration [9, 21, 22, 32, 35, 36]. However, our thesis needs further experimental validation.

Further, studies demonstrated that GFAP-positive cells may also express NeuN [24]. Darlington and co-workers [24] revealed that NeuN positive cells might also present as astrocytes harvested from primary *in vitro* neuronal culture. We did not observe any double staining

(GFAP/NeuN) in spinal cord harvested from RAGE KO SOD1 G93A and SOD1 G93A mice. Nevertheless, Zwirner et al. [25] observed GFAP positivity in neurons following traumatic brain injuries. However, the results of these studies are not clear. Darlington et al. [24] pointed to the possibility of artifacts or antibody cross-reaction during immunofluorescence staining. Further studies are required to confirm the global effect of neurodegenerative diseases on the double staining (GFAP/NeuN) of nervous tissues. We did not observe such a phenomenon in our studies.

We can speculate that the deletion of RAGE in ALS mice may inhibit progression of astroglia activation and gliosis during disease [32]. Moreover, we can hypothesize that the deletion of RAGE in ALS mice may inhibit progression of neurodegeneration and motor neuron damage (Fig 6) [7, 15, 23, 32, 37]. However, pathological mechanism of ALS is still not well known and poorly understood [1, 6, 21, 22, 35, 36, 26, 38–41].

## Conclusions

Our data demonstrated that HMGB1, S100B, CML as well as ACTB are elevated in the spinal cord harvested from ALS mice. However, deletion of RAGE decreased the level of proinflammatory RAGE ligands in ALS spinal cord. Moreover, absence of RAGE decreased the level of GFAP in ALS spinal cord and thus inhibited nucleus malfunctions as well as loss of motor neurons. We confirmed pathological effect of RAGE signaling pathway in lumbar spinal cord of ALS mice. We observed alternations in molecular pattern of proinflammatory RAGE ligands expression/amount during progression of the disease in RAGE KO SOD1 G93A mice. Our research confirmed that the neglected protein, ACTB may be an important marker in the progression of ALS in SOD1 G93A mice. However, our results need further confirmations.

## Supporting information

**S1 Fig. Immunofluorescence staining–negative control.** No primary antibodies, only secondary antibodies were used (Table 2). Images were taken under 20× objective with 0.7 numerical aperture (20× /0.7). Scale bar = 20 μm.
(TIF)

**S1 Raw images. Western blot raw data.**
(PDF)

## Acknowledgments

Authors would like to thank Staff at the Department of Histology and Embryology, University of Warmia and Mazury in Olsztyn, for letting us use the laboratory space and equipment.

## Author Contributions

**Conceptualization:** Natalia Nowicka, Kamila Zglejc-Waszak, Judyta Juranek.

**Data curation:** Judyta Juranek, Krzysztof Wąsowicz, Joanna Wojtkiewicz.

**Formal analysis:** Natalia Nowicka, Kamila Zglejc-Waszak, Agnieszka Korytko.

**Funding acquisition:** Joanna Wojtkiewicz.

**Investigation:** Kamila Zglejc-Waszak.

**Methodology:** Natalia Nowicka, Kamila Zglejc-Waszak, Agnieszka Korytko.

**Project administration:** Judyta Juranek, Agnieszka Korytko, Joanna Wojtkiewicz.

**Resources:** Judyta Juranek.

**Software:** Natalia Nowicka, Kamila Zglejc-Waszak, Agnieszka Korytko, Krzysztof Wąsowicz, Małgorzata Chmielewska-Krzesińska.

**Supervision:** Judyta Juranek, Krzysztof Wąsowicz, Joanna Wojtkiewicz.

**Validation:** Natalia Nowicka, Kamila Zglejc-Waszak, Agnieszka Korytko.

**Visualization:** Natalia Nowicka, Kamila Zglejc-Waszak, Agnieszka Korytko, Małgorzata Chmielewska-Krzesińska, Joanna Wojtkiewicz.

**Writing – original draft:** Natalia Nowicka, Kamila Zglejc-Waszak.

**Writing – review & editing:** Kamila Zglejc-Waszak, Judyta Juranek, Agnieszka Korytko.

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
