## [Decision Letter · Decision Letter 0]

10 Jan 2024

PONE-D-23-34768Novel insights into RAGE signaling pathways during the progression of amyotrophic lateral sclerosis in RAGE-deficient SOD1 G93A micePLOS ONE

Dear Dr. Zglejc-Waszak,

Thank you for submitting your manuscript to PLOS ONE. After careful consideration, we feel that it has merit but does not fully meet PLOS ONE’s publication criteria as it currently stands. Therefore, we invite you to submit a revised version of the manuscript that addresses the points raised during the review process.

We look forward to receiving your revised manuscript.

Kind regards,

Belgin Sever, Ph.D.

Academic Editor

PLOS ONE

3. To comply with PLOS ONE submissions requirements, in your Methods section, please provide additional information regarding the experiments involving animals and ensure you have included details on (1) methods of sacrifice, (2) methods of anesthesia and/or analgesia, and (3) efforts to alleviate suffering.

“This work was supported by the National Science Centre (NCN), Poland, grant No. 2017/25/B/NZ4/00435.”

“Authors would like to thank Staff at the Department of Histology and Embryology, University of Warmia and Mazury in Olsztyn, for letting us use the laboratory space and equipment. This work was supported by the National Science Centre (NCN), Poland (No. OPUS/2017/25/B/NZ4/00435).”

“This work was supported by the National Science Centre (NCN), Poland, grant No. 2017/25/B/NZ4/00435.”

7. Your ethics statement should only appear in the Methods section of your manuscript. If your ethics statement is written in any section besides the Methods, please delete it from any other section.

8. We note that Figures 1 and 6 in your submission contain copyrighted images. All PLOS content is published under the Creative Commons Attribution License (CC BY 4.0), which means that the manuscript, images, and Supporting Information files will be freely available online, and any third party is permitted to access, download, copy, distribute, and use these materials in any way, even commercially, with proper attribution. For more information, see our copyright guidelines: http://journals.plos.org/plosone/s/licenses-and-copyright.

1. You may seek permission from the original copyright holder of Figures 1 and 8 to publish the content specifically under the CC BY 4.0 license.

Reviewers' comments:

Reviewer's Responses to Questions

**Comments to the Author**

1. Is the manuscript technically sound, and do the data support the conclusions?

Reviewer #1: No

Reviewer #2: Yes

2. Has the statistical analysis been performed appropriately and rigorously? 

Reviewer #1: I Don't Know

Reviewer #2: N/A

3. Have the authors made all data underlying the findings in their manuscript fully available?

Reviewer #1: Yes

Reviewer #2: No

4. Is the manuscript presented in an intelligible fashion and written in standard English?

Reviewer #1: Yes

Reviewer #2: Yes

5. Review Comments to the Author

Reviewer #1: In the present manuscript, the authors have detailed the consequences of genetically deleting RAGE receptors in SOD1G93A mice. While the research concept is intriguing, certain matters warrant attention and resolution of the following which, once addressed these points, the manuscript should provide a more comprehensive and accurate representation of the research findings.

1) Methods Section:

- Experimental Groups and Sample Size: In the methods section, provide a detailed description of the experimental groups, including the number of mice in each group, the specific techniques employed, and the time points of assessment. Additionally, elaborate on the perfusion method used, specimen collection, and processing procedures.

- Quantification of Damaged Neuronal Nuclei: Define the criteria that characterize a damaged nucleus in the context of immunofluorescence analysis. Provide a detailed description of how this analysis was conducted, including the methods and tools used for quantification.

- Method Description for Quantification of NeuN and GFAP: In the manuscript, thoroughly describe the method used for the quantification of NeuN and GFAP in the immunofluorescence images of the spinal cord. Specify the tools and techniques employed for accurate quantification.

- NeuN Quantification; Motoneuron Counting: Elaborate on the method used to quantify NeuN and address the statement about no differences between RAGE-KO-SOD1 and SOD1 mice. Consider performing a detailed motoneuron counting to validate and support this result.

- Statistical Analysis: Statistical Analysis: Kindly elucidate the specific statistical tests and post-tests employed, along with the corresponding significance levels.

3) Results Section:

- Statistical Significance: Review and revise instances where the text suggests a correlation between increased marker levels and experimental conditions without statistically significant differences. Clearly present the statistical outcomes or reconsider the language to accurately reflect the results.

4) Additional Experiments or data:

- Microglial reaction: Include additional experiments assessing microglia reaction to the deletion of RAGE in the ALS model. Provide a comprehensive analysis to complement the findings related to NeuN and GFAP.

- Serum vs. Spinal Cord Levels of beta-Actin: Address the potential discrepancy between measuring beta-Actin levels in the serum (as cited) and the spinal cord. Discuss the proportionality of beta-Actin levels in these two contexts. Consider the feasibility of using anti-synaptophysin antibody for immunofluorescence to confirm synaptic loss.

- Numerous mice were used until reaching the terminal stage of the disease. Have the authors assessed and recorded the symptoms in these mice to identify the potential occurrence of delayed symptom onset? If so, please include such data.

6) GFAP-Positive Cells Expressing NeuN:

- Explanation: Offer a hypothesis or explanation for the observed phenomenon where GFAP-positive cells also express NeuN. Discuss potential biological implications and cite relevant literature if applicable.

7) HMGB1 Protein Activity:

Correction: Correct the statement regarding HMGB1 protein activity. Clarify that the study assessed the quantity of HMGB1 protein rather than its biological activity.

Reviewer #2: For western blot data (fig 3 and 4), recommend including a loading control for quantitative comparison of protein expression of interests, instead of normalizing based on the total protein loaded into the gel. your Western blots showed variations within the same group so better control should be used to compare protein expression levels between groups. without an adequate control, your conclusion would be weak and not fully convincing.

typo: in the abstract, there is a typo in the sentence: Moreover, inhibition of the molecular cross-talk between RAGE and its pro inflammatory ligand "my" abolish neuroinflammation.... Should be 'may'

6. PLOS authors have the option to publish the peer review history of their article (what does this mean?). If published, this will include your full peer review and any attached files.

Reviewer #1: **Yes: **Luciana Politti Cartarozzi

Reviewer #2: No

---

## [Author Response · Author response to Decision Letter 0]

2 Feb 2024

Reviewers' comments:

Reviewer #1: 

In the present manuscript, the authors have detailed the consequences of genetically deleting RAGE receptors in SOD1G93A mice. While the research concept is intriguing, certain matters warrant attention and resolution of the following which, once addressed these points, the manuscript should provide a more comprehensive and accurate representation of the research findings.

Response: We would like to thank you for the very thorough reading of our manuscript, below point-by-point response to your comments and suggestions.

Reviewer #1: 

1) Methods Section:

- Experimental Groups and Sample Size: In the methods section, provide a detailed description of the experimental groups, including the number of mice in each group, the specific techniques employed, and the time points of assessment. Additionally, elaborate on the perfusion method used, specimen collection, and processing procedures.

Response: We apologize for not being very clear. It was our oversight. We have now restructured the text in the revised version of our manuscript as per the suggestion.

Lines: 77-87, 93-94, 114, 131.

Reviewer #1:

- Quantification of Damaged Neuronal Nuclei: Define the criteria that characterize a damaged nucleus in the context of immunofluorescence analysis. Provide a detailed description of how this analysis was conducted, including the methods and tools used for quantification.

Response: Thank you for this comment. We hope that the additional information about damaged nucleus in the context of immunofluorescence analysis (lines: 138-149), has improved our manuscript.

Reviewer #1:

- Method Description for Quantification of NeuN and GFAP: In the manuscript, thoroughly describe the method used for the quantification of NeuN and GFAP in the immunofluorescence images of the spinal cord. Specify the tools and techniques employed for accurate quantification.

Response: Immunostained tissues were photographed using a fluorescence microscope (Olympus IX83) and analyzed with NIH open source Image J software. Subsequently, the areas of immunofluorescent staining were measured in four similar regions (ROI) of interest for each tissue slice (ventral horn). Four technical replicates were used per each of four biological replicates within one group. From these values, the mean area of fluorescent signal, representing the amount of GFAP and NeuN proteins and thus positive cells in each group, was calculated. 

We have now restructured the text in the revised version of our manuscript as per your suggestion. Lines: 144-149.

Reviewer #1:

- NeuN Quantification; Motoneuron Counting: Elaborate on the method used to quantify NeuN and address the statement about no differences between RAGE-KO-SOD1 and SOD1 mice. Consider performing a detailed motoneuron counting to validate and support this result.

Response: Thank you for this comment. The mean area of fluorescent signal, representing the amount of NeuN protein and thus NeuN-positive cells in each group, was calculated. We speculate that our data and hypothesis are real/true. MacLean, Juranek et al. (2021) performed quantification of overlap of NeuN+ RAGE+ area. They observed the tendency to elevated amount of NeuN (without statistical significance) in the spinal cord harvested from ALS patients with high AGER (gene encoding RAGE) compared to low AGER ALS patients. Previous results described in MacLean, Juranek et al. (2021) are similar to our present data. Hence, we assume that no differences between RAGE-KO-SOD1 and SOD1 mice are desired and proper in our studies. However, studies revealed the elevated level of NeuN/ChAT/DAPI-positive cells in the spinal cord harvested from ALS mice with AGER deletion of microglia (120th day of the disease, MacLean, et al. 2021). The slight discrepancy in the results may be due to the fact that in present MS we counted a single protein (only NeuN) and not several proteins simultaneously. Moreover, in the present studies SOD1 G93A mice were breed with RAGE knockout mice created by CRISPR-Cas9 method. We did not use tamoxifen as in MacLean et al. (2021). Additionally, we analyzed other time points (150th day of the disease) than MacLean et al. (2021). More likely, this discrepancy may be a result of the dynamic changes occurring in the spinal cord during progression of ALS. The negative control confirmed the specificity of the antibodies used during immunofluorescence staining (S1 Fig.). We are considering Cresyl Violet staining in future studies as described in the following papers (Chiarotto, Cartarozzi et al. 2019, Juranek et al. 2016). For the purposes of the review, we have performed additional analysis such as a quantification of NeuN and DAPI cells in the ALS spinal cord. Even though we followed the protocol used by MacLean et al (2021) for counting NeuN-positive cells, we did not obtain a higher amount of protein in the spinal cord harvested from ALS mice with RAGE deletion. Thus, subsequent analysis confirmed our results.

MacLean, M., Juranek, J., Cuddapah, S. et al. Microglia RAGE exacerbates the progression of neurodegeneration within the SOD1G93A murine model of amyotrophic lateral sclerosis in a sex-dependent manner. J Neuroinflammation 18, 139 (2021). https://doi.org/10.1186/s12974-021-02191-2

Juranek, J. K., Daffu, G. K., Geddis, M. S., Li, H., Rosario, R., Kaplan, B. J., Kelly, L., & Schmidt, A. M. (2016). Soluble RAGE Treatment Delays Progression of Amyotrophic Lateral Sclerosis in SOD1 Mice. Frontiers in cellular neuroscience, 10, 117. https://doi.org/10.3389/fncel.2016.00117

Chiarotto, G.B., Cartarozzi, L.P., Perez, M. et al. Tempol improves neuroinflammation and delays motor dysfunction in a mouse model (SOD1G93A) of ALS. J Neuroinflammation 16, 218 (2019). https://doi.org/10.1186/s12974-019-1598-x

Reviewer #1:

- Statistical Analysis: Kindly elucidate the specific statistical tests and post-tests employed, along with the corresponding significance levels.

Response: We apologize for the lack of clarity. It was an oversight on our part. We have now restructured the text in the revised version of our manuscript as per your suggestion. Lines: 152-156.

3) Reviewer #1:

Results Section:

- Statistical Significance: Review and revise instances where the text suggests a correlation between increased marker levels and experimental conditions without statistically significant differences. Clearly present the statistical outcomes or reconsider the language to accurately reflect the results.

Response: We apologize for the lack of clarity. This is very important information that should be included in our manuscript. We have now restructured the text in the revised version of our manuscript. As suggested, we have added information about statistical significance. Lines: 164-165, 173-174.

4) 

Reviewer #1:

Additional Experiments or data:

- Microglial reaction: Include additional experiments assessing microglia reaction to the deletion of RAGE in the ALS model. Provide a comprehensive analysis to complement the findings related to NeuN and GFAP.

Response: We have been conducting research on ALS for several years. We were members of the team that conducted the following research: Microglia RAGE exacerbates the progression of neurodegeneration within the SOD1G93A murine model of amyotrophic lateral sclerosis in a sex-dependent manner (MacLean, M., Juranek, J., Cuddapah, S. et al. 2021). The results of these studies revealed that microglia display increased RAGE immunoreactivity in the spinal cords of high AGER (gene encoding RAGE) expressing patients and in the SOD1 G93A murine model of ALS vs. respective controls. We demonstrate that microglia AGER deletion at the age of symptomatic onset, day 90, in SOD1 G93A mice extends survival in male but not female mice. Critically, many of the pathways identified in human ALS patients that accompanied increased AGER expression were significantly ameliorated by microglia AGER deletion in male SOD1 G93A mice. Results indicate that microglia RAGE disrupts communications with cell types including astrocytes and neurons, intercellular communication pathways that divert microglia from a homeostatic to an inflammatory and tissue-injurious program. Microglia RAGE contributes to the progression of SOD1 G93A murine pathology in male mice and may be relevant in human disease. 

We can speculate that the decreased amount of GFAP in the spinal cord harvested from ALS mice with RAGE gene deletion indicates that deletion/inhibition of RAGE may have therapeutic/beneficial effect in mice with ALS. MacLean, Juranek et al. (2021) revealed that male ALS mice bearing microglia AGER deficiency from the age of 3 months exhibited reduced gliosis, neuronal and motor function loss, and reduced dysfunctional transcriptomic signatures in lumbar spinal cord tissue. However, we should perform further studies to confirm our theory and results related to NeuN and GFAP proteins in the spinal cord of ALS mice with RAGE gene deletion.

In the future studies, we would like to focus on assessing microglia reaction to global deletion of RAGE in the ALS mice model. However, we have tried here to limit the scope of our MS to the abnormal RAGE mediated signaling in the ALS mouse model with the global deletion of RAGE, which help us formulate new hypothesis and scientific goals. We believe that our previous and current studies complement each other and provide a comprehensive landscape of the role of RAGE in the mouse model of ALS.

MacLean, M., Juranek, J., Cuddapah, S. et al. Microglia RAGE exacerbates the progression of neurodegeneration within the SOD1G93A murine model of amyotrophic lateral sclerosis in a sex-dependent manner. J Neuroinflammation 18, 139 (2021). https://doi.org/10.1186/s12974-021-02191-2

MacLean, M., López-Díez, R., Vasquez, C. et al. Neuronal–glial communication perturbations in murine SOD1G93A spinal cord. Commun Biol 5, 177 (2022). https://doi.org/10.1038/s42003-022-03128-y

Reviewer #1:

- Serum vs. Spinal Cord Levels of beta-Actin: Address the potential discrepancy between measuring beta-Actin levels in the serum (as cited) and the spinal cord. Discuss the proportionality of beta-Actin levels in these two contexts.

Response: Thank you for this comment. We agree with you that we should have discussed the proportionality of beta-Actin levels in the context of serum levels and the amount in the spinal cord. We have added a new information in Discussion section (lines: 299-303). Nevertheless, we have performed immunofluorescence (green color) and immunohistochemistry (brown color) staining of ACTB protein in spinal cord harvested from mice with ALS and control adult mice. Our studies confirmed expression of ACTB in spinal cord (ventral horn).

Moreover, cytoskeleton genes or proteins are neglected molecules in ALS studies. Kubinski and Claus (2022) revealed, that proteins involved in ALS are connected with actin cytoskeletal proteins, therein: ACTA1. We examined the expression of ACTA1 mRNA in mice model of ALS. We observed that in the spinal cord the expression of ACTA1 mRNA was increased in the terminal group of SOD1 mice compared to the control group (WT) (P=0.024). Our data showed that actin molecules, ACTA1, ACTB, may be putative targets in ALS diagnosis and therapy. In the future studies, we would like to focus on the actin cytoskeleton protein in the spinal cord harvested from ALS mice. 

Kubinski S, Claus P. Protein Network Analysis Reveals a Functional Connectivity of Dysregulated Processes in ALS and SMA. Neurosci Insights. 2022;17:26331055221087740 

Reviewer #1:

Consider the feasibility of using anti-synaptophysin antibody for immunofluorescence to confirm synaptic loss.

Response: The studies revealed that ALS is caused by the progressive degeneration of motor neurons in the spinal cord which causes a reduction in the number of motor neuron synapses (Tomiyama, Cartarozzi et al. 2023). Davis (2008) revealed decreased amount of synaptophysin (SYP) protein in spinal cord harvested from ALS model of mice compared to control group. However, Tomiyama, Cartarozzi et al. (2023) revealed that the amount of SYP in the lumbar ventral horn of ALS mice was maintained during IFNβ treatment. Hence, we can speculate that in RAGE KO SOD1 mouse the amount of SYP in the ventral horn of the spinal cord may be similar to that in the control group (WT).

Moreover, Davis (2008) revealed that the amount of VAChT was similar in ALS mice and control mice at the terminal stage of experiment. However, Davis (2008) in his thesis observed that width of VAChT-immunoreactive synapses of ALS mice were elevated than control group at terminal stage. For the purposes of the review, we have performed immunohistochemical analysis vesicular acetylcholine transporter (VAChT) protein in order to cholinergic synapse frequency. 

We analyzed the amount of VAChT protein in the spinal cord harvested from mice in the terminal groups, i.e. WT and SOD1 (150 days of ALS). Based on the amount of protein, we can estimate the number of synapses in the spinal cord ventral horns during ALS. Immunohistochemistry revealed the amount of VAChT protein in spinal cord harvested from ALS model of mice was similar with control group of mice (WT). Moreover, our data and Davis (2008) results confirm synaptic VAChT-immunoreactive in terminal stage ALS mice. However, in future we plan to expand the studies and check the amount of VAChT as well as SYP proteins in spinal cord harvested from ALS mice with RAGE deletion. We will remember to perform these analyzes in the future studies. 

Davis, K; (2008) Evidence of disease resistance in cholinergic synapses on spinal motor neurons in a mouse model of amyotrophic lateral sclerosis. Doctoral thesis , UCL (University College London) https://discovery.ucl.ac.uk/id/eprint/1568415

Tomiyama ALMR, Cartarozzi LP, de Oliveira Coser L, Chiarotto GB, Oliveira ALR. Neuroprotection by upregulation of the major histocompatibility complex class I (MHC I) in SOD1G93A mice. Front Cell Neurosci. 2023 Aug 30;17:1211486. doi: 10.3389/fncel.2023.1211486

Reviewer #1:

- Numerous mice were used until reaching the terminal stage of the disease. Have the authors assessed and recorded the symptoms in these mice to identify the potential occurrence of delayed symptom onset? If so, please include such data.

Response: The role of RAGE in the progression of ALS is well documented but still not fully understood. The presented publication is a continuation of our many years of research on the role of RAGE signaling in the mouse model of ALS (Nowicka et al. 2022, 2019, Juranek et al. 2022, 2016, 2015). As described in Nowicka et al. (2022) experimental and control animals were randomly divided into studied groups per defined time points: 90 and the terminal stage of the disease (about 150 days, primary endpoint, natural demise). All mice were carefully weighted starting from the eighth week of age. The weight was recorded three times per week. Mice from two time point groups, i.e. 90 and 150, were sacrificed independently at each of these time points regardless of their health status. The onset of the disease was determined when observed weight loss. RAGE KO SOD1 G93A mice and controls were sacrificed at the same time points as SOD1 G93A mice. Results from motor function test, i.e. results of hanging cage test are deposited in the earlier publication (Nowicka et al. 2022). Moreover, results from Kaplan–Meier curves showing survival probability of experimental terminal group as well as Life Quality, Mobility and Lifespan of Mice with Congenital ALS are described in (Nowicka et al. 2022). Lee and co-workers (2020) revealed that absence of RAGE increased survival as well as muscle strength in the SOD1 G93A mouse model of ALS. In the presented studies, we continued our previous assumptions and hypothesis, that RAGE signaling pathway plays a crucial role in progression of ALS in mice. The aim of the present study is to demonstrate the molecular effect of inhibiting RAGE signaling in SOD1 G93A transgenic mice. Therefore, in this study, we focused exclusively on RAGE signaling pathways and thus on the protein abundance. Please, respect our approach and research methods. 

Reviewer #1:

6) GFAP-Positive Cells Expressing NeuN:

- Explanation: Offer a hypothesis or explanation for the observed phenomenon where GFAP-positive cells also express NeuN. Discuss potential biological implications and cite relevant literature if applicable.

Response: This is a very interesting question. The studies indicated that GFAP is one of the major intermediate filament proteins in mature astrocytes. NeuN is a widely expressed labels in nuclei of mature neurons. However, we fully agree with your hypothesis that GFAP-positive cells of spinal cord may also express NeuN. Darlington et al. (2008) revealed that NeuN is expressed not only in motor neuron nuclei but also in primary astrocytes from mouse, rat and human brain. Moreover, Darlington et al (2008) indicated that NeuN may label the nuclei of astrocytes cultured in vitro from human fetus, human adult, newborn rat, and embryonic mouse brain. Unlike these findings in vitro, but consistent with previous studies in vivo, astrocytes in sections of postnatal day 10 rat brain did not express NeuN (Darlington et al 2008). However, we did not find any information about double staining (GFAP/NeuN) of the spinal cord harvested from adult mice (Tomiyama, Cartarozzi et al. 2023). Therefore, we assume that we do not observe positive double staining (GFAP/NeuN) of the spinal cord in adult mice. In summary, we hypothesize/assume that the NeuN antibody selectively stains mice neurons in vivo, i.e. in a tissue section. However, the label of NeuN in in vitro experiments is more widespread that the specification states. When staining cells from in vitro cultures, we should remember about the possibility of double staining (GFAP/NeuN) of astrocytes (Darlington et al 2008). 

With this in mind, experiments with staining neuronal cells harvested from in vitro primary neuronal cultures should be additionally stained with another neuronal marker such as MAP2. 

However, we assume that positive double staining (GFAP/NeuN) of the spinal cord in adult mice may take place during traumatic spinal cord injury. The similar phenomenon takes place during traumatic brain injury in Alzheimer patients (Zwirner et al. 2021). However, further studies revealed that this phenomenon may be an artefact or potential neuronal immunopositivity for GFAP either cross-reactivity of antibodies. Further studies are necessary to confirm the double staining theory (GFAP/NeuN) as a consequence of traumatic spinal cord/brain injury or even peripheral injury (Victório, Cartarozzi et al. 2012). We decided to add information about this phenomenon in the discussion section. Lines: 318-325.

Darlington, P. J., Goldman, J. S., Cui, Q. L., Antel, J. P., & Kennedy, T. E. (2008). Widespread immunoreactivity for neuronal nuclei in cultured human and rodent astrocytes. Journal of neurochemistry, 104(5), 1201–1209. https://doi.org/10.1111/j.1471-4159.2007.05043.x

Tomiyama ALMR, Cartarozzi LP, de Oliveira Coser L, Chiarotto GB, Oliveira ALR. Neuroprotection by upregulation of the major histocompatibility complex class I (MHC I) in SOD1G93A mice. Front Cell Neurosci. 2023 Aug 30;17:1211486. doi: 10.3389/fncel.2023.1211486

Zwirner, Johann et al. “GFAP positivity in neurons following traumatic brain injuries.” International journal of legal medicine vol. 135,6 (2021): 2323-2333. doi:10.1007/s00414-021-02568-1

Victório SC, Cartarozzi LP, Hell RC, Oliveira AL. Decreased MHC I expression in IFN γ mutant mice alters synaptic elimination in the spinal cord after peripheral injury. J Neuroinflammation. 2012 May 7;9:88. doi: 10.1186/1742-2094-9-88

Reviewer #1:

7) HMGB1 Protein Activity: Correction: Correct the statement regarding HMGB1 protein activity. Clarify that the study assessed the quantity of HMGB1 protein rather than its biological activity.

Response: We apologize for not being very clear. We have now restructured the text in the revised version of our manuscript as per your suggestion.

Reviewer #2: 1. For western blot data (fig 3 and 4), recommend including a loading control for quantitative comparison of protein expression of interests, instead of normalizing based on the total protein loaded into the gel. your Western blots showed variations within the same group so better control should be used to compare protein expression levels between groups. without an adequate control, your conclusion would be weak and not fully convincing.

Response: We thank you for noting that our manuscript has generated new and potentially important information about the mouse model of ALS. We appreciate the investment of time such careful reviews take. Please find our point-by-point response below.

Western blot (WB) is an excellent molecular technique for determining protein expression. In our previous papers (Zglejc-Waszak et al. 2023, Nowicka et al. 2022, Jaroslawska et al. 2021), we used total protein analysis as a reliable loading control for quantitative fluorescent Western blotting. We choose this method of analysis because we study protein expression during neurodegenerative diseases. To our knowledge, common controls, including β-actin, GAPDH and tubulin, are differentially expressed in tissues from a wide range of animal models of neurodegeneration (Aldridge et al. 2008, Eaton et al. 2013). We demonstrated variable expression of β-actin in the spinal cord harvested from mice from the study groups (Fig. 3 and Fig. 4). Moreover, Eaton et al. (2013) revealed that expression of these “control” proteins was not consistent between different portions of the same tissue, highlighting the importance of careful and consistent tissue sampling for WB experiments. Therefore, normalizing WB data to control for a protein such as β-actin in such circumstances may result in ‘skewing’ of all data in ALS research. Consequently, we propose it would be prudent to use total protein analysis to save time, resources, increase sensitivity and accuracy as well as the working range of protein load for quantitative Western blotting. In our opinion, total protein analysis should be considered as an alternative reference standard for data normalization in modern quantitative fluorescent Western blotting, particularly during neurodegenerative studies where control expression may be variable (https://www.bio-rad.com/en-pl/applications-technologies/total-protein-normalization?ID=PODYJQRT8IG9). Stain-Free Western Blotting does not require the housekeeping gene because all calculations were performed automatically by the software including the Normalization Factor and Normalized Volume. The target protein band intensity values were adjusted for variation in the protein load. This method allowed for accurate comparison of target proteins among the samples (https://www.bio-rad.com/webroot/web/pdf/lsr/literature/Bulletin_6390.pdf). Please, respect our approach and research methods. In our opinion, choosing total protein analysis as a reliable loading control for quantitative fluorescent Western blotting is the best choice.

Aldridge GM, Podrebarac DM, Greenough WT, Weiler IJ. The use of total protein stains as loading controls: an alternative to high-abundance single-protein controls in semi-quantitative immunoblotting. J Neurosci Methods. 2008;172(2):250-254. doi:10.1016/j.jneumeth.2008.05.003 

Eaton SL, Roche SL, Llavero Hurtado M, et al. Total protein analysis as a reliable loading control for quantitative fluorescent Western blotting. PLoS One. 2013;8(8):e72457. Published 2013 Aug 30. doi:10.1371/journal.pone.0072457 

Jaroslawska J, Korytko A, Zglejc-Waszak K, et al. Peripheral Neuropathy Presents Similar Symptoms and Pathological Changes in Both High-Fat Diet and Pharmacologically Induced Pre- and Diabetic Mouse Models. Life (Basel). 2021;11(11):1267. Published 2021 Nov 19. doi:10.3390/life11111267 

Nowicka N, Szymańska K, Juranek J, et al. The Involvement of RAGE and Its Ligands during Progression of ALS in SOD1 G93A Transgenic Mice. Int J Mol Sci. 2022;23(4):2184. Published 2022 Feb 16. doi:10.3390/ijms23042184 

Zglejc-Waszak K, Mukherjee K, Korytko A, et al. Novel insights into the nervous system affected by prolonged hyperglycemia. J Mol Med (Berl). 2023;101(8):1015-1028. doi:10.1007/s00109-023-02347- y

Reviewer #2: 2. typo: in the abstract, there is a typo in the sentence: Moreover, inhibition of the molecular cross-talk between RAGE and its pro inflammatory ligand "my" abolish neuroinflammation.... Should be 'may'

Response: Thank you for this comment. We have corrected our typing mistake.

---

## [Editor Report · Decision Letter 1]

13 Feb 2024

Novel insights into RAGE signaling pathways during the progression of amyotrophic lateral sclerosis in RAGE-deficient SOD1 G93A mice

PONE-D-23-34768R1

Dear Dr. Zglejc-Waszak,

We’re pleased to inform you that your manuscript has been judged scientifically suitable for publication and will be formally accepted for publication once it meets all outstanding technical requirements.

Kind regards,

Belgin Sever, Ph.D.

Academic Editor

PLOS ONE
---

## [Editor Report · Acceptance letter]

29 Feb 2024

PONE-D-23-34768R1 

PLOS ONE

Dear Dr. Zglejc-Waszak, 

I'm pleased to inform you that your manuscript has been deemed suitable for publication in PLOS ONE. Congratulations! Your manuscript is now being handed over to our production team.

Kind regards, 

on behalf of

Assoc. Prof. Dr. Belgin Sever 

Academic Editor

PLOS ONE